# Low-Dose Estrogens as Neuroendocrine Modulators in Functional Hypothalamic Amenorrhea (FHA): The Putative Triggering of the Positive Feedback Mechanism(s)

**DOI:** 10.3390/biomedicines11061763

**Published:** 2023-06-20

**Authors:** Christian Battipaglia, Tabatha Petrillo, Elisa Semprini, Francesco Ricciardiello, Maria Laura Rusce, Greta Prampolini, Fedora Ambrosetti, Alessandra Sponzilli, Alessandro D. Genazzani

**Affiliations:** Center for Gynecological Endocrinology, Department of Obstetrics and Gynecology, University of Modena and Reggio Emilia, Via del Pozzo 41, 41100 Modena, Italy; christianbattipaglia@gmail.com (C.B.); tabatha.petrillo@gmail.com (T.P.); elisasemprini93@gmail.com (E.S.); francesco.ricciardiello96@gmail.com (F.R.); mlaura.rusce@gmail.com (M.L.R.); greta.prampolini@icloud.com (G.P.); fedora.ambrosetti01@universitadipavia.it (F.A.); alessandra.sponzilli@gmail.com (A.S.)

**Keywords:** functional hypothalamic amenorrhea, opioids, estriol, estradiol, positive feedback, stress, anovulation

## Abstract

Functional hypothalamic amenorrhea (FHA) is a non-organic reversible chronic endocrine disorder characterized by an impaired pulsatile secretion of the gonadotropin-releasing hormone (GnRH) from the hypothalamus. This impaired secretion, triggered by psychosocial and metabolic stressors, leads to an abnormal pituitary production of gonadotropins. As LH and FSH release is defective, the ovarian function is steadily reduced, inducing a systemic hypoestrogenic condition characterized by amenorrhea, vaginal atrophy, mood changes and increased risk of osteoporosis and cardiovascular disease. Diagnosis of FHA is made excluding other possible causes for secondary amenorrhea, and it is based upon the findings of low serum gonadotropins and estradiol (E2) with evidence of precipitating factors (excessive exercise, low weight, stress). Treatments of women with FHA include weight gain through an appropriate diet and physical activity reduction, psychological support, and integrative approach up to estrogen replacement therapy. If no spontaneous ovarian function is restored, assisted reproductive technologies may be used when pregnancy is desired. Because subjects with FHA are hypoestrogenic, the use of low-dose estrogens has been proposed as a putative treatment to positively modulate the spontaneous restart of gonadotropin secretion, counteracting the blockade of the reproductive axis triggered by stress acting through the neuroendocrine pathways at the basis of positive feedback of estrogens. The mechanism through which low-dose estrogens acts is still unknown, but kisspeptin-secreting neurons may be involved.

## 1. Introduction

Functional hypothalamic amenorrhea (FHA) is a reversible endocrine disorder characterized by a disturbance of the pulsatile secretion in the hypothalamus of the gonadotropin-releasing hormone (GnRH), which results in the impaired function of the hypothalamic–pituitary–ovarian axis, chronic anovulation and hypoestrogenism [1]. It is responsible for approximately 25–35% and 3% of secondary and primary amenorrhea cases, respectively [2].

Risk factors for FHA include low-weight eating disorders (up to anorexia nervosa) and other causes, such as a low body weight due to excessive exercise and stress [3]. FHA often occurs in female athletes. Indeed the combination of low energy availability, hypothalamic–pituitary–gonadal axis inhibition resulting in menstrual dysfunction, and low bone density is called the “female athlete triad” [4].

Psychosocial and metabolic stressors, including eating disorders and excessive exercise, induce various hormonal changes that impair hypothalamic secretion of GnRH, leading to an abnormal pituitary production of gonadotropins and to the failure of follicle recruitment, thus leading to anovulation and hypoestrogenism [5].

GnRH–luteinizing hormone (LH) disturbances in FHA include a wide range of features: both a lower and higher mean frequency of LH pulses, complete absence of LH pulsatility and normal-appearing secretion pattern [6]. The reduction in GnRH drive results in LH and FSH levels that are too low to stimulate full folliculogenesis and ovulatory ovarian function [7].

All these changes in FHA patients may be considered a natural protective mechanism that temporarily suppresses reproductive functions when physical conditions are not suitable to sustaining a pregnancy [8].

In FHA, as a consequence of energy deficiency from under-nutrition or from excessive energy use, a cascade of energy-conservation mechanisms take place, such as reduction of the concentrations of glucose, insulin, leptin, insulin-like growth factor (IGF-1) and kisspeptin (Kp), or the elevation of growth hormone (GH), neuropeptide Y, ghrelin, beta-endorphin and cortisol plasma levels [9].

The occurrence of a hypoestrogenic condition in FHA negatively affects most of the estrogen-sensitive organs. It is best to resolve this within a reasonable span of time (a few months), as otherwise fertility issues as well as an increased risk of osteoporosis and cardiovascular disease might occur. In addition, women with FHA may experience symptoms such as vaginal atrophy and mood changes [10]. 

On this basis, this review aims to provide recent insights with regard to low-dose estrogen administration and their effects on neuroendocrine impairments in patients with FHA.

## 2. Diagnosis of FHA

Functional hypothalamic amenorrhea should be differentiated from other forms of primary or secondary amenorrhea. The diagnosis of FHA typically involves a combination of a medical history, physical examination and laboratory tests. The diagnostic process should also consider other potential causes of amenorrhea, such as pregnancy, thyroid dysfunction, or polycystic ovary syndrome (PCOS) [11]. Diagnosis of FHA is based upon the findings of some clinical and laboratory issues, such as amenorrhea, low serum gonadotropins and estradiol (E2), with evidence of a precipitating factor (exercise, low weight, stress).

Before the onset of amenorrhea, women with FHA have normal cycles, which become irregular and then cease after loss of weight, increased exercise activity or significant stress occur together or one after another. In some women, menses may stop without a preceding period of oligomenorrhea. If a patient had exercise-induced amenorrhea in the past that remitted when she decreased exercising, it is likely to recur if she resumes exercising without a compensatory caloric increase [7].

When a hypogonadotropic hypogonadism is evidenced, the key diagnostic tool is a GnRH stimulation test, which in the case of FHA shows a positive response of the gonadotropins to exogenous GnRH [12]. Such a test is relevant for a correct diagnosis in those cases where amenorrhea occurred just after menarche, thus permitting one to differentiate a delayed puberty from a FHA.

The administration of a progestin such as dihydrogesterone for 10 days, usually named as “progestin challenge test”, may be useful, since FHA is associated with scant or no withdrawal bleeding as E2 levels are deficient. In conditions such as PCOS, in which E2 levels are only relatively low, bleeding usually occurs following the administration of exogenous progestin. An imaging study to assess the internal genitalia may be of value in adolescents with primary amenorrhea before the progestin challenge test [7].

Once the hypothalamic origin has been found, it is important to rule out genetic diseases, such as Kallman syndrome (characterized by anosmia) or Prader–Willi syndrome (with characteristic hyperphagia, obesity and retardation). Features such as delayed puberty, primary amenorrhea and the presence of additional symptoms (anosmia, mental retardation, extreme obesity, facial dysmorphia, malabsorption) are suggestive of congenital diseases [13]. Imaging evaluation should be performed to exclude organic diseases of the hypothalamic area (neoplasms, tuberculosis, parasitosis, sarcoidosis) [11].

If amenorrhea is present for more than six months, or earlier in case of a suspicion of severe nutritional deficiency, other energy deficit states or a history of fragility fractures, a baseline bone mineral density (BMD) should be performed [14].

## 3. Treatment of Functional Hypothalamic Amenorrhea

Treatment of women with FHA includes treatment of the underlying cause of hypogonadotropic hypogonadism (energy deficit from insufficient caloric intake, excessive exercise or emotional stress) and its consequences, such as low BMD, anovulatory oligo-amenorrhea, and infertility or genitourinary symptoms (vaginal dryness and dyspareunia) due to estrogen deficiency [7].

Patient treatment for women with FHA should consider whether severe bradycardia, hypotension, orthostasis, and/or electrolyte imbalance is present [15].

Low energy availability leads to hypothalamic–pituitary–ovarian (HPO) axis disruption as a defensive mechanism to save energy, as reflected in menstrual dysfunction. Energy availability can be described as the difference between energy intake and exercise energy expenditure, normalized to fat-free mass [4]. Weight gain through refeeding and improved energy availability in amenorrhoeic patients with anorexia nervosa correlated with the resumption of menses [16]. The approach should include dietary evaluation and counselling as well as psychological support for treating stress and enhancing behavioral change [17].

It is important to note that amenorrhea may persist for some time after the reversal of precipitating factors and that at least up to 6 to 12 months of weight stabilization may be required for the resumption of menses. Golden et al. suggested that a weight gain of 2.0 kg more than the weight at which menses stopped was needed for the restoration of menses [18].

In some cases, even after weight stabilization, regular menses never resume. This suggests that factors other than nutrition play a key role in FHA pathophysiology, such as stress or other psychological disorders. In comparison to eumenorrheic women, women with FHA have been observed to exhibit a higher incidence of dysfunctional attitudes, struggle to cope with daily stresses and have a history of mental health issues and mood disorders [19]. In these patients, psychological support, such as cognitive behavioral therapy (CBT), shows the capacity to restore ovarian function and also alter metabolic function, improving cortisol, leptin, and TSH [20].

Estrogen replacement therapy may be considered after 6 to 12 months of nutritional, psychological, and exercise-related interventions in those with a low bone density and/or evidence of skeletal fragility. 

The Endocrine Society’s 2017 Clinical Practice Guideline for the diagnosis and treatment of FHA advises against the use of oral contraceptive pills (OCPs) for the sole purpose of regaining menses or improving BMD, since OCPs may hide bone loss not being so effective on bone mass deposition [7]. 

The use of transdermal E2 therapy with cyclic oral progestin should be preferred over oral hormonal therapies, as it has been demonstrated to improve lumbar and hip BMD in patients with FHA [21]. Transdermal estrogen likely has a more positive effect on BMD than OCPs because it does not affect IGF-I secretion, a bone-trophic hormone that is downregulated by OCPs [22,23].

The use of bisphosphonates, denosumab, testosterone, and leptin to improve BMD in adolescents and women with FHA is not recommended [7].

In patients with FHA wishing to conceive, after a complete fertility work-up, induction of ovulation should be attained. As a first-line treatment, pulsatile GnRH followed by gonadotropin therapy should be chosen [7].

In most patients with FHA, exogenous GnRH or exogenous gonadotropin would likely be efficacious for inducing ovulation, as pituitary–ovarian feedback mechanisms are intact. This technique leads to more physiologic ovulatory menstrual cycles with monofollicular development and minimal risk of developing multiple pregnancies [24,25], (Figure 1).

Up to now, there have been no randomized clinical trials that have evaluated the use of clomiphene citrate, an estrogen receptor antagonist, in inducing ovulation and treating infertility in women with FHA. In fact, these patients are characterized by low estrogen levels, and they do not show the activation of the estrogen-induced negative feedback. In addition, Djurovic et al. reported that after 10 days of treatment with clomiphene citrate, menstrual bleeding occurred in only 9 out of 17 patients who recovered a normal body weight but not in those who showed a diagnosis of anorexia nervosa [26]. Therefore, treatment with clomiphene citrate may be considered if a woman has sufficient endogenous estrogen levels [27].

A BMI of at least 18.5 kg/m^2^ is considered to be the minimal threshold that a woman needs to optimize her chances for fertility. Moreover, an extremely low BMI is associated with a higher risk of adverse pregnancy outcomes [28]. Therefore, induction of ovulation should be limited to women with a satisfactory body weight.

AMH plasma levels can be used to estimate ovarian reserve in women with hypothalamic hypogonadism [29], as gonadotropins will be low and the identification of primary ovarian insufficiency (POI) may be delayed due to the fact that FHA induces a reduced gonadotropin output [29].

## 4. Other Treatments: The Role of Low-Dose Estrogens

Hypoestrogenism and low luteinizing hormone (LH) plasma levels are the main endocrine abnormalities observed in patients with FHA [12]. Because subjects with FHA are hypoestrogenic, the use of low-dose estrogens has been proposed as a putative treatment. Such an approach was thought to be effective on the basis of the positive feedback exerted by estrogens centrally during the follicular phase of the menstrual cycle [30], when such a feedback is considered essential to strengthen the effects of FSH and LH on granulosa and theca cells [5].

In fact, one of the first studies investigating the role of low-dose estrogens was conducted in 1978 [31]. The effects of the use of epimestrol, a synthetic steroidal estrogen, administered at a dose of 5 mg every 6 h for 5 days, was able to improve basal levels of LH, FSH, prolactin (PRL), estradiol (E2), progesterone (PRG), dehydroepiandrosterone sulfate (DHEA-S) and also the response of gonadotropins to LH-releasing hormone (LH-RH) and of TSH to thyrotropin-releasing hormone (TRH) stimulation in 18 women with secondary amenorrhea and oligomenorrhea of hypothalamic-pituitary origin. The pituitary response to LH-RH indicated a substantially more pronounced LH secretion than before treatment, while all three hormone levels rapidly declined to values close to basal levels twelve hours after therapy was stopped. A second test was performed 36 h after the last drug administration, showing a significantly higher LH response than the one found under basal conditions, while no other significant variations were observed in the FSH response to LH-RH, nor in the PRL response to TRH [31].

Later, in a 2012 study, our group [32] evaluated the effects of the short-term administration of estriol in women with FHA. Twelve patients with a mean age of 27.5 ± 1.2 years, with amenorrhea for at least 6 months and LH plasma levels <3 mIU/mL, were selected to participate in the study. Their body weight was stable for the 6 months prior to the study, and BMI was not below 19 kg/m^2^. Women with psychiatric diseases, intense training for agonistic purposes and adrenal, thyroid, or prolactin (PRL) diseases were not included. All patients were asked to make no changes in their lifestyle and were evaluated for endocrine tests, such as GnRH test, and basal hormone parameters before and after 8 weeks of estriol administration (2 mg/day orally). A statistically significant increase in LH plasma levels and in the LH dynamics of secretion was observed after the treatment interval, while no changes occurred in the other hormonal parameters. The GnRH-induced LH response showed a higher amplitude after estriol administration than in baseline conditions.

The ultrasound evaluation did not demonstrate any changes of endometrial thickness after the treatment interval, and none of the patients resumed their menstrual cycle during the duration of the treatment.

Such observations were in agreement with previous observations conducted with epimestrol in 1978 [31]; as a matter of fact, both epimestrol and estriol were demonstrated to positively affect the release of gonadotropins from the pituitary as well as its response to GnRH stimulation. 

The administration of weak estrogens is capable of increasing both gonadotropins’ plasma levels and the amount of gonadotropin secreted after a GnRH bolus while not changing estradiol plasma concentrations, supporting the hypothesis that the conversion to estradiol was minimal in both studies [31,32]. The changes after epimestrol administration were also showed in other studies conducted by different groups on women and also on men [33,34]. Based on this, it can be argued that low-dose estrogens reactivate hypothalamic–pituitary control, probably both inducing a greater expression/function of the receptor for GnRH and improving the synthesis/production of LH at the gonadotropic cell level.

Though no changes of spontaneous LH pulse frequency occurred, the LH pulse amplitude was increased after estriol administration, and GnRH-induced LH secretion showed a higher amplitude [32]. Similar changes were seen in a previous study in animal models in which estriol was reported to increase spontaneous and GnRH-induced LH release [35] but also in a more recent study conducted by our group on women with Kallmann syndrome [36]. 

To stress the fact that weak estrogens, as estriols are, are rapidly effective at the pituitary level, our group investigated short-term estriol effects by administering 2 mg/day of estriol to twelve patients with FHA for just 10 days. The concentrations of plasma LH and FSH were significantly increased by day 10 of treatment, and a significant increase in their responses to the GnRH bolus was also evidenced [37], thus demonstrating that gonadotropin secretion could be modulated and induced after just a short interval of treatment, though with lower increases than after 8 weeks of estriol administration [32].

Conversely to what was observed under short-term epimestrol administration [31] and long-term estriol administration [32], short-term use of estriol for 10 days also changed E2, Androstenedione, and TSH plasma levels. Since cortisol was not changed, the adrenal gland was considered to not be affected by estriol administration. This suggested that, in this case, androstenedione was mainly of ovarian origin. It can be argued that such an elevation of Androstenedione may reflect an increase of gonadotropin plasma levels responsible for a higher stimulation of granulosa cells with a higher Androstenedione production, later converted by aromatase into E2 [38]. The increase of E2 plasma levels was not reported under long-term estriol administration, but this might be due to the fact that for the 2016 study [37], a more sensitive immunoassay sensitivity was used.

As for the reduction of TSH plasma levels under short-term estriol administration [37], it is important to remark that patients with FHA usually show normal TSH levels, while fT3 might be reduced (the so-called “low T3 syndrome”), with no changes under E2 treatment [39]. The TSH plasma level changes after estriol treatment might be a consequence similar to what was observed in ovariectomized and stressed animals, in which a low dose of E2 administration decreased the TRH-mRNA content in the hypothalamus [40].

Recently, the effects of a very low estradiol (E2) dose, as low as 5 ng every day for 12 weeks, was tested, evaluating the response of LH to a GnRH test and of LH and cortisol to a naloxone bolus [41]. The authors reported that no other changes occurred in the hormonal parameters, including estradiol; however, both LH and FSH plasma levels considerably rose. Out of 17 individuals, only 2 reported menstrual bleeding within the interval of treatment. In this study, the Instantaneous Secretory Rates (ISR) Computation was used in order to evaluate the LH secretory rate of the LH-induced response to the GnRH bolus [42]. Both in terms of the plasma concentration and computing ISR, the LH response to the GnRH bolus resulted in being greater than before treatment.

When the Naloxone test was evaluated, both LH and cortisol responses were considered. Before the treatment, LH showed a response 30 to 60 min after the bolus, while after estradiol administration, the LH response to the naloxone bolus occurred after just 15 min. Cortisol plasma levels were in the upper range of normal values, as is often reported in patients with FHA [43], and did not change after estradiol administration. However, while no response to Naloxone was observed for cortisol under baseline conditions, after estradiol treatment, naloxone infusion was able to significantly increase cortisol plasma levels within 30 min after the bolus, resulting in a higher cortisol response than that observed under baseline conditions.

These results support the hypothesis that very low estradiol amounts, as low as those of estriol [32,37], were able to trigger the biological effects and response due to the positive feedback, mediated by estrogens, at the pituitary level. 

Recent reports have shown that Kp-secreting neurons play a significant role in the regulation of the reproductive axis because their secretion is controlled by the positive or negative feedback produced by gonadal hormones [44].

Moreover, the presence of a cortisol response to naloxone infusion observed after low-dose estradiol administration supports the hypothesis that a specific neuroendocrine modulation took place on the Adreno Corticotropin Hormone (ACTH) pathway [45]. 

The very low estradiol amounts supplied presumably modified or adapted the naloxone’s ability to bind opioidergic receptors, probably also acting on the modulation and/or expression and synthesis of opioidergic receptors. This may have caused the cortisol response to occur [41].

More specifically, low-dose estradiol probably led to an increased level of opioid receptors’ expression and synthesis while acting with a positive feedback on kisspeptin-induced GnRH release, allowing naloxone to bind the receptors more competitively and displace endogenous opioids such as β-endorphins (EP) more efficiently.

The statistically significant increase in the amplitude of spontaneous LH secretory pulses shown by all studies with low-dose estrogen administration has also been observed when using either neuroactive drugs such as naltrexone [46,47] or acetyl-L-carnitine [48,49], confirming that several kinds of molecules can affect the hypothalamus-pituitary function modulating both GnRH secretion and gonadotropes’ ability to synthesize/store/release LH.

The mechanism through which low-dose estrogens induce a central positive feedback is still unknown, but kisspeptin-secreting neurons may be involved. Kisspeptin/neurokinin B/dynorphin (KNDy) neurons are located in the infundibular region of the hypothalamus, in the arcuate nucleus (ARC), and express different receptors, also including estradiol α receptors (ERα) [50]. Kisspeptin (Kp) is also produced by a second group of neurons located in the anteroventral periventricular nucleus (AVPV) of the hypothalamus; these neurons are devoid of neurokinin-B and dynorphin production and mediate the positive feedback from estrogens [50]. 

In the normal ovarian cycle, low blood levels of estradiol (E2) and progesterone mainly activate kisspeptin neurons located in the AVPV, increasing the production of Kp, while higher levels of sex steroid hormones modulate KNDy neurons’ function in the ARC, exerting an inhibitory function on Kp secretion [50]. Since Kp acts directly on GnRH neurons, modulating the pulsatile release of GnRH, low-dose estrogens may induce a central positive feedback acting primarily on AVPV neurons modulating Kp production [1,50,51,52] (Figure 2).

In conclusion, all these findings suggest that in patients with FHA, while neither contraception nor hormone replacement therapy (HRT) are adequate for positively modulating/triggering the spontaneous restart of gonadotropin secretion, weak estrogens as well as hyper-low-dose estradiol seem to be more effective and may play an important role in counteracting the defensive neuroendocrine mechanisms triggered by stress that block the reproductive axis.

## Figures and Tables

**Figure 1 biomedicines-11-01763-f001:**
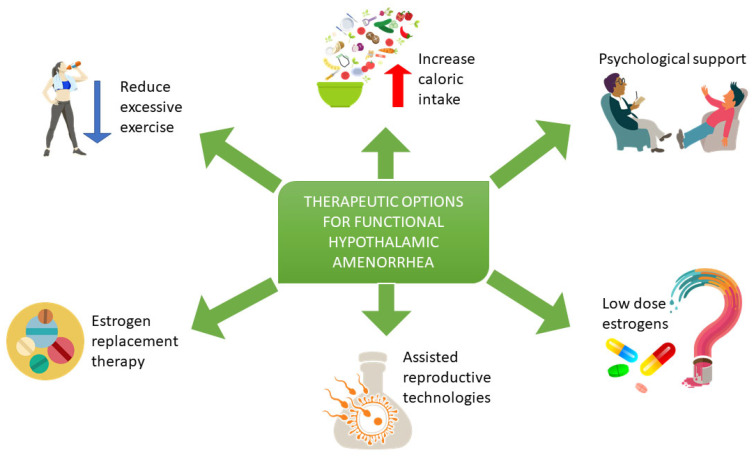
Treatment of women with FHA includes a reduction of excessive exercise, dietary evaluation and psychological support to reduce stress, an enhancement of behavioral change and an increase of energy availability. Estrogen replacement therapy may be considered after 6 to 12 months of nutritional, psychological, and exercise-related interventions in those with low bone density and/or evidence of skeletal fragility. Assisted reproductive technologies may be considered in patients wishing to conceive. Low-dose estrogen use in patients with FHA is still under study.

**Figure 2 biomedicines-11-01763-f002:**
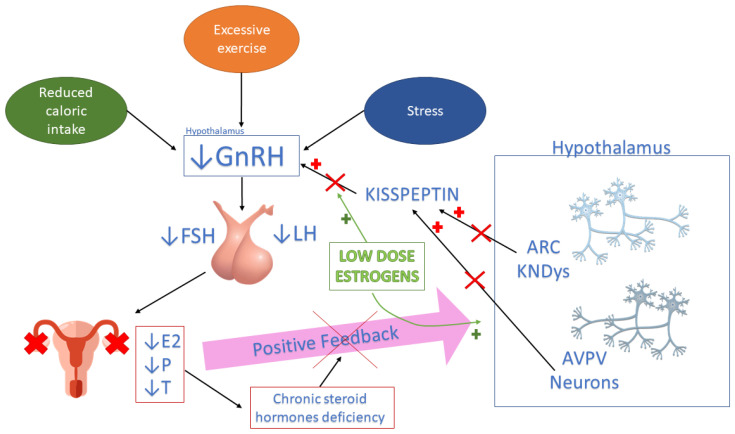
Psychosocial and metabolic factors, such as reduced caloric intake, excessive exercise and stress, impair hypothalamic secretion of GnRH, leading to an abnormal pituitary production of gonadotropins and to anovulation and hypoestrogenism. The chronic lack of steroid hormone production inhibits the positive feedback usually exerted by estrogen levels on KNDys and AVPV neurons, blocking ovarian function. Low-dose estrogens may induce a central positive feedback acting primarily on AVPV neurons modulating kisspeptin production and consequently stimulating GnRH pituitary secretion. E2—estradiol; P—progesterone; T—testosterone; KNDys—Kisspeptin, neurokinin B, and dynorphin (KNDy) neurons; AVPV—anteroventral periventricular nucleus; ARC—arcuate nucleus.

## Data Availability

Not applicable.

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
