# Peer review of "Low-Dose Estrogens as Neuroendocrine Modulators in Functional Hypothalamic Amenorrhea (FHA): The Putative Triggering of the Positive Feedback Mechanism(s)"

_biomedicines, 2023, doi:10.3390/biomedicines11061763_

Round 1
Reviewer 1 Report
The manuscript titled Low dose or weak estrogens administration trigger a neuroendocrine change in functional hypothalamic amenorrhea (FHA) through positive feedback mechansisms by Battipaglia and co-workers reviewed the literature and presented evidence of this phenomena. The title is a bit of a reach and is not supported by the referenced studies. This has been hypothesized, but all that has been observed is a positive response of the gonadotropins to low dose estrogens (some without statistical power). That this is truly working through a positive feedback mechanism—such as hypothalamic kisspeptin is not established in the cited publications. Suggest tempering the title.
Other considerations:
Line 22 – Is should not be capitalized “…function is restored, assisted... may be used [when] pregnancy is desired.”
Line 40. Run on sentence. Suggest break at “…female athletes. Indeed the combination…”
Line 61. Suggest “has to be” should be “best to be resolved…”
Line 85. Briefly describe the “progestin challenge”
Line 92. I think it should be “hyper-orexia” or “hyperphagia”. I believe hyperphagia is a more common term.
Line 123- 125. Awkward sentence. Reword for clarity. Maybe “This suggests factors other than nutrition play a key role in FHA pathophysiology such as stress or other psychological disorders.”
Line 147 – 149. This needs a citation.
Line 175. Suggest placing the citation (29) following “AMH measurements (29)”
Line 187. Add the citation
Line 220. “…estriol [were] demonstrated…”
Line 239. Since most readers would be unfamiliar with the meaning of “2 mg/die” suggest “2 mg/day”
Line 240-241. Suggest “Concentration of plasma LH and FSH were increased by day 10 of treatment”
Line 248. Suggest not abbreviating androstenedione as “A”—it does read well as an abbreviation—just spell it out for clarity.
Line 256. Suggest ending the sentence at “..higher sensitivity.”
Line 256. Confused as to what a “naloxone bolus of LH” means. Was “LH and cortisol were measured in response to a naloxone bolus”? rephrase for clarity.
Line 293. The suggestion that opoid receptors were induced by low dose estradiol needs support of a citation.
Line 309 -316. This paragraph needs some citations
The manuscript needs some editing for clarity, but overall language use is acceptable.
Author Response
Reviewer # 1
Changes have highlighted in azure color
The comment is correct. For this reason the title has been modified accordingly
Line 22 – The commenti s correct. We have changed the sentence as suggested.
Line 40 – Changes have been done as suggested.
Line 61 – Changed as suggested
Line 85 – The paragraph has been implemented and definition of progestin challange test has been added
Line 92 – As suggested hyperorexia has been substituted with hyperphagia
Line 123-125 – The sentence has been implemented as suggested.
Line 149 – A reference has been added as requested.
Line 175 – Reference has been added as requested
Line 187 – Citation has been added as requested.
Line 220 – The sentence has been changed as suggested
Line 2239 – Changes has been done as suggested
Line 240 – The sentence has been modified as suggested.
Line 248 – “A” has been spelled
Line 257 – The sentence has been changed as requested.
Line 265 – The sentence has been rephrased as suggested to improve clarity
Line 293 – The commenti s correct. A reference has been added.
Line 309 – Specific new reference has been added as requested

Reviewer 2 Report
Overall, it is an interesting review. However, I consider the authors need to improve some sections and overall structure to make the manuscript ready for publication.
First, the title is misleading; it appears as if the manuscript is an actual experiment rather than a review paper. Please rephrase.
Second, there's a general issue with the paragraph structure of the manuscript. There are sections with multiple small paragraphs that should be combined together because they are all related to the same idea/concept. Examples; Concepts from paragraph Ln 186- Ln 192 should be fused/combined with the following paragraph. Same with paragraphs among these lines; 208-218. There are multiple instances like these throughout the manuscript (for example Ln 317-325). Please revise accordingly in all instances.
There's also an issue with paragraphs that include concepts/ideas that are not connected with the rest of the manuscript. This creates an issue with the reading flow. Please revise as well. Example; Ln 96-101; both paragraphs are isolated ideas without connection, which affects the flow. Same for lines 134-139. Also, paragraph 162 starts by mentioning clomiphene citrate without any background or previous definition of it. Again, these examples occur multiple times throughout the manuscript, please revise in all instances. Focus mostly on sections where there are multiple short (1-2 sentences) paragraphs.
There's no objective stated for this review. Why did the authors choose to do a review on this topic? Why is this topic important to review? Please add in the introduction section.
Please consider subdividing the current sections into subsections. Some sections are too long (e.g., section 4) and thus having subsections might help improve the flow.
Conclusion (Ln 303) should be the last paragraph.
Figure 2; explain acronyms in legend (e.g., "P", "T", etc.).
English grammar looks acceptable throughout the manuscript.
Author Response
Reviewer 2
Changes have been highlithed in green
As suggested the title has been changed accordingly
As suggested the paragraphs line 186-192 have been merged and fused together
As suggested the paragraphs line 208-218 have been merged and fused together
As suggested the paragraphs line 317-325 have been modified trying to maintain the specific sense about the role of Kisspeptin in being actor of the positive feedback driving CnRH secretion
Line 96-101 – The phrases have been changed and merged as suggested
Line 134-139 has been changed as suggested
Line 161-163 has been changed as suggested
As requested, a specific statment was added at the end of the introduction about the aim of the review.
The final paragraph and the conclusions have changed as suggested, merging the final paragraphs
Fig 2 – Acronyms have been explained in the legend

Round 2
Reviewer 2 Report
The authors did a good job editing the manuscript and it is now ready for publication.
Minor edits required